# High Temperature Adiabatic Heating in µ-IM Mould Cavities—A Case for Venting Design Solutions

**DOI:** 10.3390/mi11040358

**Published:** 2020-03-30

**Authors:** Matthew Tucker, Christian A. Griffiths, Andrew Rees, Gethin Llewelyn

**Affiliations:** College of Engineering, Swansea University, Swansea SA1 8EN, UK; 709390@Swansea.ac.uk (M.T.); Andrew.Rees@Swansea.ac.uk (A.R.); 656688@swansea.ac.uk (G.L.)

**Keywords:** micro injection moulding, adiabatic heating, diesel effect, venting

## Abstract

Micro-injection moulding (µ-IM) is a fabrication method that is used to produce miniature parts on a mass production scale. This work investigates how the process parameter settings result in adiabatic heating from gas trapped and rapidly compressed within the mould cavity. The heating of the resident air can result in the diesel effect within the cavity and this can degrade the polymer part in production and lead to damage of the mould. The study uses Autodesk Moldflow to simulate the process and identify accurate boundary conditions to be used in a gas law model to generate an informed prediction of temperatures within the moulding cavity. The results are then compared to physical experiments using the same processing parameters. Findings from the study show that without venting extreme temperature conditions can be present during the filling stage of the process and that venting solutions should be considered when using µ-IM.

## 1. Introduction

The demand for micro parts has significantly increased due to the ability of the Micro-injection Moulding (µ-IM) manufacturing process to produce a wide variety of components [1,2]. Different µ-IM machines are appearing within the market and these machines can now meet the high accuracy and dimensional requirements demanded by the consumers [3,4]. There is an ever increasing demand for the production of large quantities of micro parts and µ-IM can meet the technical requirements of these products [5]. 

Previous studies in µ-IM have demonstrated that varying processing conditions such as mould and melt temperature, injection speed and air evacuation in the mould cavity can all have an effect on the resulting process outputs. A study by Griffiths et al., showed that that barrel temperature and injection speed are the key factors that influence the aspect ratios of micro features replicated in Polypropylene (PP) and Acrylonitrile butadiene styrene (ABS) [5]. Further studies have concluded that in µ-IM a high melt and mould temperature along with high injection speeds are required to enhance micro component replication fidelity [6,7,8]. Mönkkönen et al. found that in addition to the polymer used the most influential parameters were the injection speed, melt temperature and holding pressure [9], whereas others such as Shen et al., found that the most influential parameter was having a mould temperature above the polymer glass transition temperature was optimum for consistent replication of micro injected parts [10]. In a study by Yu et al., the quality of filling as a function of distance from the gate concluded that cavities towards the end of the melt flow filled to a higher yield [8]. The general consensus from the literature is that the high processing setting benefit the µ-IM process [5,6,7,8,9,10]. Whereas the increase in the process setting of higher temperatures and injection speed is often seen as an optimisation solution for µ-IM, it is clear that there are also negative effects to these process settings. In particular, an increase in the melt and mould temperature increases the process cycle time as extra cooling time is required [5]. 

During the filling process, the polymer has a specific volume (V), which varies depending on both the resultant pressure (P) and temperature (T). *PVT* data is used to represent the melt flows compressibility and shows the functional dependence between the polymers volume, pressure and temperature [11]. When cavity air temperature is kept at a constant heat for a given volume, the temperature will only rise when the air is further compressed resulting in work performed on the system. This increase in temperature is referred to as the adiabatic temperature and the whole procedure is referred to as adiabatic heating. Furthermore, as the temperature is increased there is an increase in the resulting pressure which will continue to increase the more it is compressed [12]. For the adiabatic heat process, the ideal gas law outlines the relationship between the air Pressure (P), Volume (V), Temperature (T)and the number of moles (n) of an ideal gas. For an adiabatic process, there is no heat transfer that takes place, meaning no heat is added or removed to the system. An ideal gas has a given number of atoms in a specified volume and when this changes it inversely effects the pressure and linearly the temperature. This is expressed in Equation (1) below where R is the value for the universal gas constant.
(1)PV=nRT
where for an adiabatic process the final temperature *T*_2_ is given as:(2)$$T_{2=T_{1}{\left(\frac{V_{1}}{V_{2}}\right)}^{\gamma -1}}$$

In addition, the final pressure *P*_2_ is given as:(3)$$P_{2=P_{1}{\left(\frac{V_{1}}{V_{2}}\right)}^{\gamma }}$$
where *T_1_*, *P_1_* and *V_1_* are the initial state values and *T_2_*, *P_2_* and *V_2_* are the final state values. γ is the ratio of heat capacity at constant pressure (*C_p_*) to heat capacity at constant volume (*C_v_*) [13]. Air is a diatomic gas made up of around 78% Nitrogen, 20% Oxygen, 0.9% Argon and the further 1.1% made up of additional elements and has a specific heat ratio given as γ=1.4 [13]. 

The adiabatic process can result in extremely high temperatures within the µ-IM mould, with the potential to cause ignition of the compressed gases. This combustion is referred to as the diesel effect, which can lead to damage of the mould cavity as well as degradation of the polymer part [7]. Like the cycle of a diesel engine, the polymer is injected at high speed where it rapidly compresses the air within the mould cavity, which in turns leads the temperature to adiabatically increase. 

This high-pressure environment within a mould cavity combined with volatile gases chemically released from the polymer can increase the likelihood of ignition as the auto ignition temperature of the air and gas mixture is achieved. For ignition to occur, there must be a stoichiometric mixture of oxygen and process gasses within the cavity [14]. Process issues due to the combustion in the mould are typically burn marks, short shots, poor surface finish or a change in structural property of the polymer [15] 

During the µ-IM process failure to integrate air evacuation within the cycle has the potential to introduce poor process control and damage the mould tooling [12]. When the polymer melt is injected into the mould cavity, the flow front pushes the unvented air towards the end of the cavity causing it to compress. This compression increases the air temperature significantly within the mould. It can result in outgassing, decomposition of mould compounds or leave corrosive residue [12]. Within conventional injection moulding, it is found that the resident air will be evacuated out of the mould cavity through the mould parting line and coarse grain grinding of the surfaces can be used to facilitate this [16]. Passive venting can also be provided with gaps machined at the parting line surfaces (usually at the end of a flow front). These permissible vents allow air to escape without significant pressure build. Usually they are 1.5–2.5 mm wide and the vent depth depends on the polymer being processed. PP, Polyamide (PA), Polyoxymethylene (POM), Polyethylene (PE) use a <15 µm depth vent and Polystyrene (PS), ABS, Polycarbonate (PC), polymethylmethacrylate (PMMA) use a <30 µm depth vent [16]. In addition to this, the pressurised flow front can push air into ejector pins. Venting pins can be made if the pin is 20–50 µm smaller than the bore for a length of 300 µm [16]. 

Often the dimensions of these vents are larger than some micro parts, so macro venting design rules do not easily translate to micro moulds. Yao and Kim identified that components manufactured by µ-IM fall into one of the following two categories. Type A are components with overall sizes of less than 1 mm while Type B have larger overall dimensions but incorporate micro features with sizes typically smaller than 200 μm [17]. So it can be seen that due to scaling issues some of the rules for macro venting solutions cannot be used in µ-IM. Ideally, the primary vent is present at the split line of the mould faces but with high precision surface achieved when manufacturing moulds using micro machining processes [18] there can be insufficient gaps between the split lines. 

Currently, changes to the processing conditions and altering the injection locations and injection speed profiles is used to prevent air traps but with very fast injection times this method has limitations. Also, the majority of micro parts are considered to be ‘blind-holes’ where the air gets trapped and gets compressed at high flow speeds and this cause resistance to the melt flow resulting in abnormalities within the final part production such as uneven flow fronts [19]. 

To aid the evacuation of air from a mould cavity, vacuum venting has been introduced within µ-IM tooling platforms. This feature aids in the counteracting pressure that is being produced by the compressed air at the flow front which improves replication accuracy and process control [20]. Utilising vacuum venting results in the ability to reduce the flow resistance of the polymer melt that would usually be effected by trapped air within the cavity or micro features [21]. In the study by Lucchetta et al., the importance of ensuring that assisted venting does not reduce the temperature of the mould surface was highlighted for temperature sensitive polymers [22]. 

This research will demonstrate the requirement for venting design solutions when considering mass manufacture of polymer micro components. A micro test part will be produced using the µ-IM process to establish if adiabatic heating will occur from gas trapped and rapidly compressed within the mould cavity. In the methodology section, the Autodesk Moldflow (Autodesk, San Rafael, CA, USA) setup for simulating part manufacture with varying processing parameters using the Design of Experiments (DOE) method is shown. This is then followed by the application of an adiabatic heating model using the boundary conditions established from the simulation to generate an informed prediction of temperatures within the moulding cavity. Finally, the adiabatic heating and diesel effect results are presented and compared to parts produced with the same process settings used in the simulation. 

## 2. Experimental Procedure

### 2.1. Part Design

For this study, a geometry (Figure 1) was used which is suitable for replication using a Battenfeld Microsystem 50 µ-IM machine (Wittmann Battenfeld GmbH, Kottingbrunn, Austria). The geometry had a constant thickness of 0.5 mm, a runner length of 40 mm and a final rectangular section measuring 15 mm × 5 mm. Each of the corners had a radius of 0.5 mm in order to reduce shear. The geometry had an overall surface area of 249.75 mm^2^ and a volume of 47.61 mm^3^. The total flow length of the part was 55.8 mm (Table 1).

### 2.2. Moldflow Simulation

In this study, the simulation software Autodesk Moldflow Insight 2018 was used. A study by Xie et al. concluded that a 2.5D meshes such as the Moldflow dual domain mesh do not capture all effects taking place [23]. Therefore, a full 3D meshwais used. Following preliminary mesh sensitivity analysis an element size of 0.1 mm was used as illustrated in Figure 2. 

### 2.3. Materials

The material properties for PP and ABS are displayed in Table 2. Both materials are used extensively within an industrial context [24]. 

### 2.4. Boundary Conditions

The simulation boundary conditions and process settings were the same as those used when processing PP and ABS on the Battenfeld 50 µ-IM machine. The mould geometry was set as solid walls whilst the tooling material was P-20 tool steel. Table 3 below shows the mechanical and thermal properties of the steel. One injection gate was used and is displayed in Figure 2. Table 4 illustrates the process variables used for the simulation L9 DOE. 

## 3. Results

### 3.1. Simulation 

#### 3.1.1. Moldflow Air Trap Results

Air traps are formed when the flow front of the polymer melt compresses air against the cavity wall when there is insufficient venting in place. The simulations identify regions of the test part where air is trapped and where there is potential for the diesel effect. For PP all nine tests successfully filled the mould and all contain air traps on the end wall of the cavity. Figure 3 shows an air trap witnessed in test one and this is a typical representation from all performed experiments. The result is informative in that it shows the part and tool designer where potential part quality issues may arise due to gassing and where vents need to be considered. When processing with ABS only three tests which utilised the higher melt temperatures (Test 3, 6 and 9) achieved a completely filled cavity. For the filled parts the air trap positions are similar to those observed for the PP simulations.

#### 3.1.2. Moldflow PP and ABS Temperature Results

Figure 4 displays the resulting Flow Front Temperatures (*T_ff_*) reached in the simulations when compared against the input Temperatures (*T_i_*) detailed in Table 4. For both materials an increase in *T_ff_* temperature from the *T_i_* is observed. Test 4 had the largest increase in temperatures from the *T_i_.* In particular, for PP and ABS an increase in 8.13% and 19.5% respectively was witnessed. 

Typically, polymers have an absolute maximum melt temperature. Any increase in this value can result in degradation of the polymer. For the PP test 3 experiment, the absolute maximum melt temperature for the material (280 °C) is exceeded by 2.6 °C due to a shear heating temperature in the cavity. This means that this setting should be avoided and that there is also potential for increased gassing from polymer degradation. For both materials it can be seen that the process window based on the four factors has a large influence on the temperature of the polymer within the cavity and this has the potential to influence the temperature of the resident air within the cavity.

#### 3.1.3. Moldflow PP and ABS Shear Rate Results

In Figure 5, the simulation shear rate results for PP and ABS are presented. For both materials, tests three, four and eight had the largest presence of shear within their cycles. These three tests utilised the highest injection speed of 800 mm/s. In comparison tests one, five and nine achieved the lowest presence of shear having been processed at the lowest injection speeds of 200 mm/s. The findings demonstrate that the process parameters have a major influence on the resulting shear but importantly no single test goes above the critical shear rate for PP (1 × 10^5^ 1/S) and ABS (5 × 10^4^ 1/S) [25]. The results demonstrate the potential of shear to increase the temperature and contribute to the formation of gasses within the moulding cavity. 

#### 3.1.4. PP Analysis of Individual Tests

Further analysis has been performed using the results presented in Figure 4. In particular, test three where the highest bulk and *T_ff_* have been observed. Bulk temperature is used, as it is too difficult to display the temperature change of the polymer within one display as the temperature profile is dynamic and changes with time, location and thickness during the injection process. It also represents the energy transported through particular locations in which it has more physical significance than average temperature. The *T_ff_* value is obtained via the fill analysis within Moldflow and represents the polymer temperature as it reaches a specified point. The original melt temperature *T_i_* at the point of injection for test three was 270 °C (*T_i_*) and it increased to 286 °C (*T_ff_*) for the bulk temperature. Also, the flow front temperature had an increase from 270 °C (*T_i_*) to 282.6 °C (*T_ff_*). Within the Moldflow analysis the largest region of temperature increase is within the runner of the moulded part where the polymer is experiencing shear, then when it flows into the main cavity it starts to cool. Moldflow does not account for adiabatic heating of the resident air within the model hence no further temperature increases are observed. In reality the flow conditions within the mould presents an ideal environment for the diesel effect to occur whereby the resident air would see a rapid rise in initial temperature before adiabatically compressed. 

#### 3.1.5. ABS Analysis of Individual Tests

As with the comparison of the temperature results for PP, test three will be used as this has the highest process parameter conditions and is the most likely test that would see the diesel effect occur. The original melt temperature *T_i_* at the point of injection was 280 °C (*T_i_*) and it increased to 303.4 °C (*T_ff_*) for the bulk temperature. Also, the flow front temperature increased from 280 °C (*T_i_*) to 302.7 °C (T*_ff_*). This temperature rise is the consequence of shear heating as observed in Figure 5. 

### 3.2. Adiabatic Heating

#### 3.2.1. Adiabatic Conditions for PP

As the Moldflow software does not account for compressional heat rise in the model, further analysis is required to understand the temperature and shear rate simulation results. Using the results of the *T_ff_* from Moldflow together with the ideal gas law for adiabatic heating (Equation (1)), it is possible to estimate the temperature of the compressed air within the mould cavity. Figure 6 shows the pressure conditions that lead to adiabatic heating occurring for both the PP and ABS polymers. It can be seen that the pressure increases in accordance to the reduction of the volume within the mould cavity. In particular of major relevance for µ-IM is that when the unfilled cavity volume is reduced beyond 5 mm^3^ the pressure gradients are altered significantly and there is an exponential increase of pressure in the mould cavity. This shows that without venting of the air within the cavity there is an adiabatic process that is taking place, which can result in diesel effect occurring within the mould. 

If the temperature of the resident air within the cavity is already equal to the mould temperature, it can be expected that the resident air will heat up rapidly when in contact with the flow front temperature. Without any venting, the rapid compression of this heated air during the filling process will result in a diesel effect where temperatures within the mould will rise to above 1300 °C (Figure 7). The model calculates that the resident air is compressed to around 0.5 mm^3^, which simulates an air trap in the mould cavity. According to the ideal gas law the smaller the volume that is achieved under compression the higher the final temperature and it is expected that combustion will occur before such extreme temperatures are reached. At high temperature gassing has already occurred in which fumes are released from the chemical properties of the polymers; this is potentially harmful to the mould and localised etching can be expected. Ignition of the resident gas can take place above 357 °C, as this is the flashpoint of PP as marked by the blue dashed line on the graph in Figure 7. Above this line the risk of damage to the part and the mould remain high. As shown, combustion will occur when the resident air is compressed to around 15 mm^3^. However, it can be seen that the air-gas mixture has the potential to reach higher temperatures as the volume of air is reduced. 

The results in Figure 7 display extremely high temperatures that without adequate venting could be experienced for all nine tests. This can cause extreme problems for the mould cavity itself. To demonstrate this, Figure 7 also displays a red horizontal dashed line indicating the melting temperature of the P20 steel at 1426 °C. The continuous cyclic temperature rise and fall has the potential of causing damage to the mould steel. The potential localised microstructure change could compromise the tooling integrity as typically clamping forces for the mould can reach 50 kN [26]. Previous research studies have demonstrated that venting has proven to reduce the resulting cavity pressure [27]. In addition, studies have demonstrated that when processing PP above the recommended melt temperatures of 180 °C to 240 °C in non-vented moulds there can be severe chemical degradation within the polymer part brought on by elevated temperatures [28].

#### 3.2.2. Adiabatic Conditions for ABS

Similar to the results for PP, the ABS polymer also undergoes adiabatic heating according to the ideal gas law. Assuming that the polymer *T_ff_* increases the resident air temperature, Figure 8 shows the temperature profile when adiabatic heating occurs. All test results show a resulting cavity temperature above 1300 °C. It is noted that the *T_ff_* received from the ABS results in a higher adiabatic temperature increase when compared to PP. In Figure 8 the red dashed lines indicates the melt point temperature of 1426 °C for the P20 tool steel mould material [29]. With the addition of adiabatic temperature rises, mould temperatures can exceed the auto ignition temperature of ABS which is between 500 °C and 575 °C and represented by the blue dashed line in Figure 8. This demonstrates that without venting all experiments can produce the diesel effect and lead to part and mould damage. 

### 3.3. Comparison with Experimental Data

In order to validate the simulation results, the experimental work that Griffiths et al. presented is used. This work uses the same part (Figure 1) and process settings (Table 4) used for the simulation results. 180 test runs were performed and the part flow lengths of ten parts for each process combination were measured. For the simulations the Moldflow measurement tool is used to measure the part flow length and it is calculated from the beginning of the runner to the end of the flow front. The physical parts produced had irregular flow fronts so the difference between the highest and lowest portion of the flow front was measured for each part. 

It was found that the tests with the highest process parameters had the largest flow lengths (Table 5) [30]. The experimental tests revealed that when comparing the two polymers, PP had the highest average flow length for all nine of the tests conducted. Not all of the ABS parts filled so it was not possible to produce a trapped air and diesel effect in the cavity, therefore these experiments are not considered. For the PP physical experiments, tests three, six and nine had the highest melt temperature and achieved the best filling results (Figure 9). Figure 9 also shows the end of fill images imposed on the CAD model for test one, seven and three. Test three had the highest flow length and this was achieved with high melt and mould temperatures and the highest injection speed of 800 mm/s. This supports the simulation results, which show that these tests can reach the highest *T_ff_* (Figure 4). However, the simulation could not show the trapped air pocket as the simulation assumes the mould is fully vented and therefore no air is trapped. 

The adiabatic model used in this research proves that it is possible to produce a diesel effect if the mould used does not have sufficient venting. The physical experiments also show that the diesel effect is present. The PP results show that the experiment with the lowest flow length (Table 5) and the lowest *T_ff_* (Figure 4) was test 1. It can be seen in Figure 10 that the part has a rounded flow front as expected for a part that is unfilled. Because the part is unfilled there is no possibility of an excessive gas trap. For test three in Figure 9 there is clear evidence of a gas trap. This experiment has the highest flow length and the highest *T_ff_* and as seen in the figure the flow front is uneven and uncharacteristic of a normal flow front. On further inspection of test part from this experiment it can be seen that there are gas pockets within the parts and there is evidence of polymer damage (Figure 11). The PP test parts produced with a high melt temperature displayed signs of part damage due to the resident air inside the cavity, this result confirms the high temperatures observed in the simulations.

## 4. Conclusions

This paper presents an investigation on the influence on µ-IM process parameter settings on adiabatic heating from gas trapped and rapidly compressed within the mould cavity. In the study PP and ABS polymers were processed with a range of different parameters to identify boundary conditions for use within a gas law model. The model is then used to identify adiabatic conditions within the mould cavity. The conclusions are as follows:Autodesk Moldflow simulations of an established part design can predict accurate temperature distributions within the µ-IM process. These results can then be used to identify accurate boundary conditions to be used in the gas law model to generate an informed prediction of temperature increases within the moulding cavity.In a mould with limited venting, extreme temperature conditions can be present during the filling stage of the process. The results show the maximum air temperature while processing PP can exceed 1300 °C when the melt flow front temperatures are between 238–283 °C. When processing ABS material the mould temperature can exceed 1400 °C when the melt flow front temperatures are above 270 °C. With such significant temperature increases it is highly likely that the polymer parts will degrade and the tooling will experience damage with prolonged use.Further work should consider improving the model by the addition of the heat transfer rate of the polymer flow front temperature to the resident air within the mould cavity. The influence of different polymer gases during processing should also be considered for their contribution to the diesel effect.The simulation of the factors that influence temperature together with the gas model highlight the potential for adiabatic heating and the physical experiments show that gas traps and part damage are experienced with combinations of process settings. The model shows extreme temperatures within the cavity, the highest temperatures are unlikely to arise as there will always be some natural venting. However, it also shows that with limited venting there is a temperature increase that is detrimental to the process. Due to size limitations macro mould venting solutions cannot always be considered for micro moulds and the findings highlight the need for designs that consider novel venting and air evacuation solutions for improved part quality and tool life.

## Figures and Tables

**Figure 1 micromachines-11-00358-f001:**
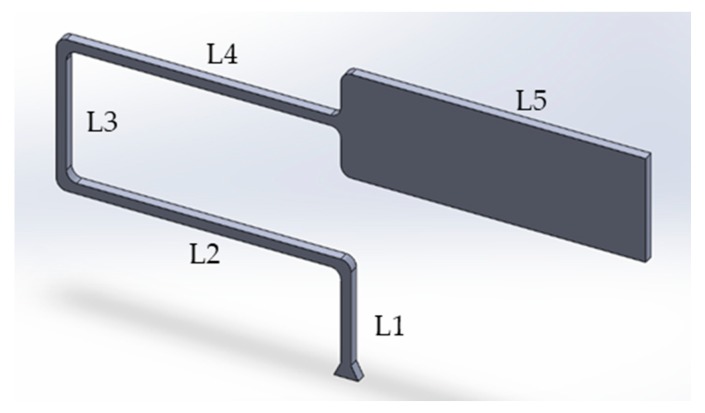
Test part geometry.

**Figure 2 micromachines-11-00358-f002:**
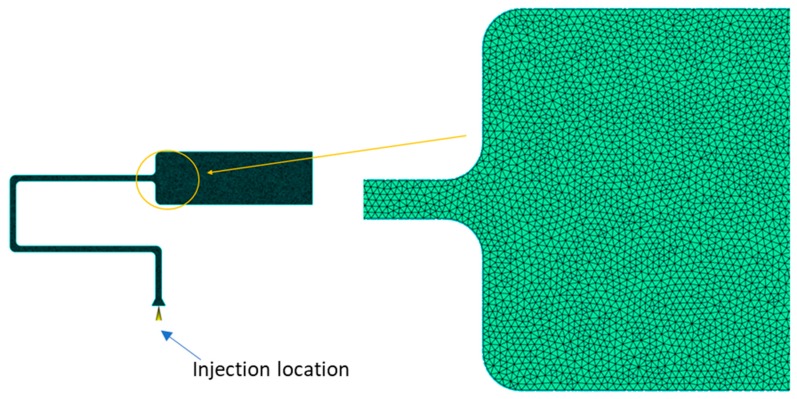
Element size 0.1mm 3D (Fine mesh).

**Figure 3 micromachines-11-00358-f003:**
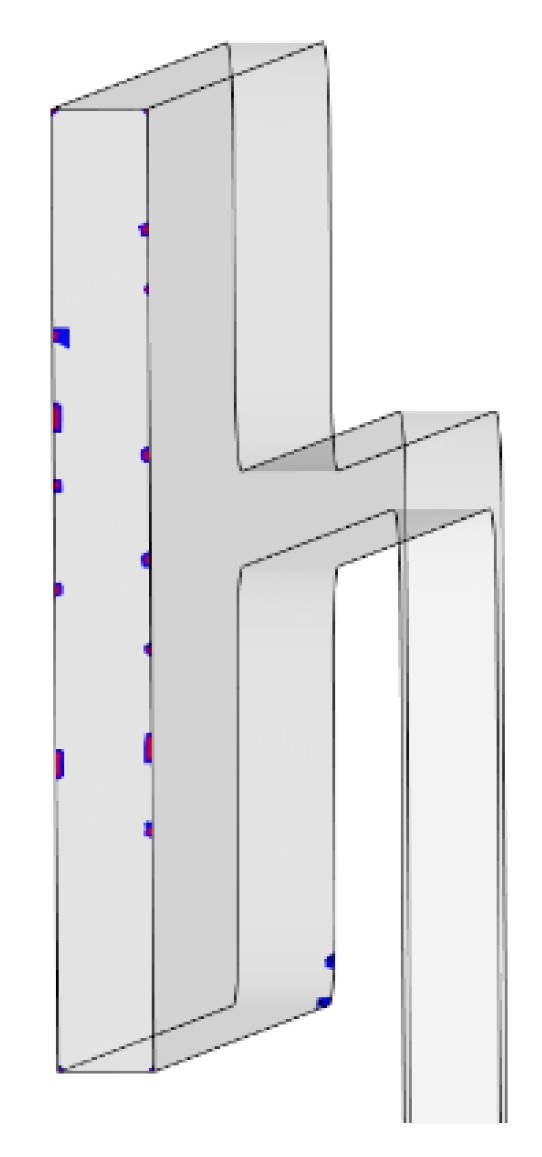
Air traps witnessed in Polypropylene (PP) simulations.

**Figure 4 micromachines-11-00358-f004:**
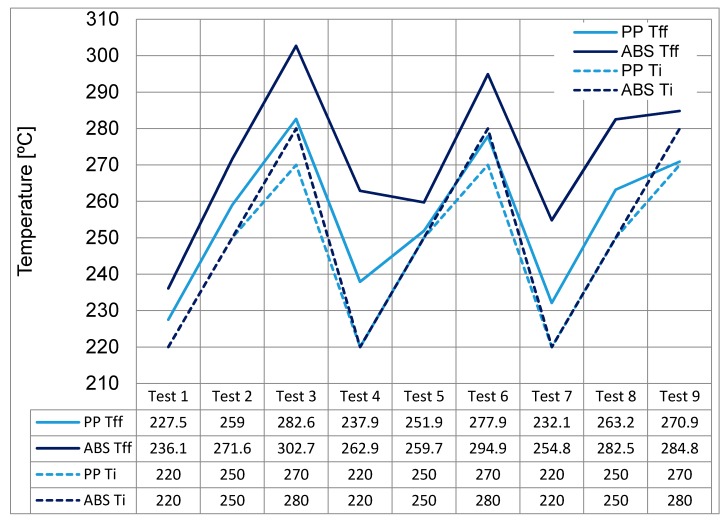
Comparison between *T_i_* and *T_ff_* on filling for PP and Acrylonitrile butadiene styrene (ABS).

**Figure 5 micromachines-11-00358-f005:**
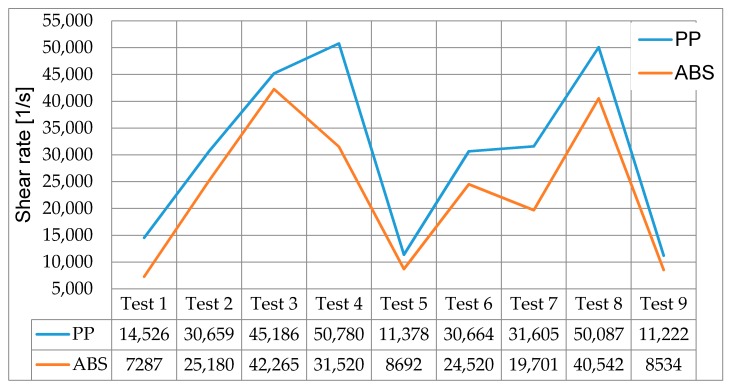
Maximum shear rate for PP and ABS.

**Figure 6 micromachines-11-00358-f006:**
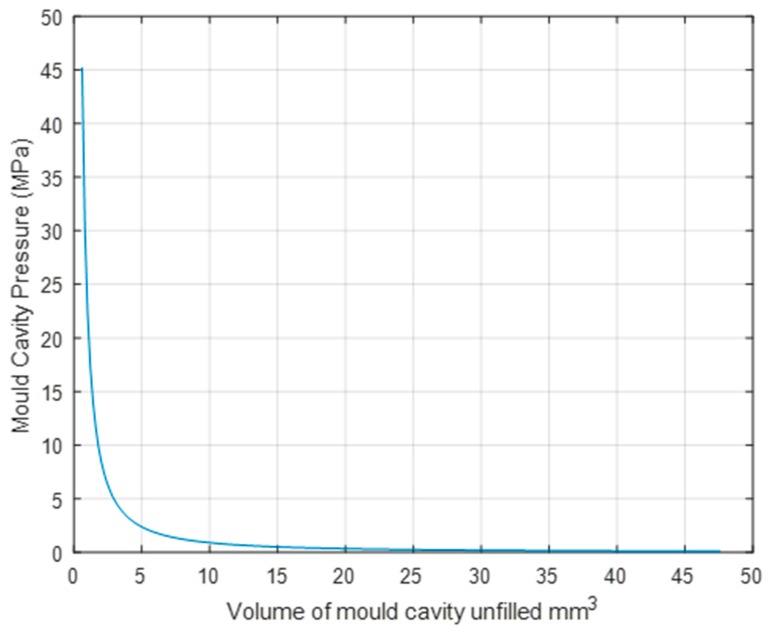
Pressure increase when air is compressed in mould.

**Figure 7 micromachines-11-00358-f007:**
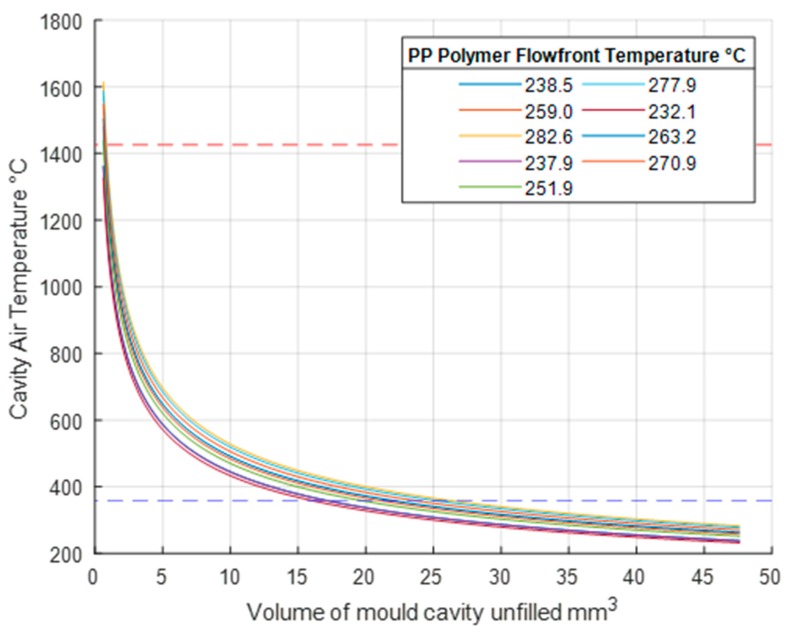
Adiabatic temperature increase when air is compressed in mould, PP.

**Figure 8 micromachines-11-00358-f008:**
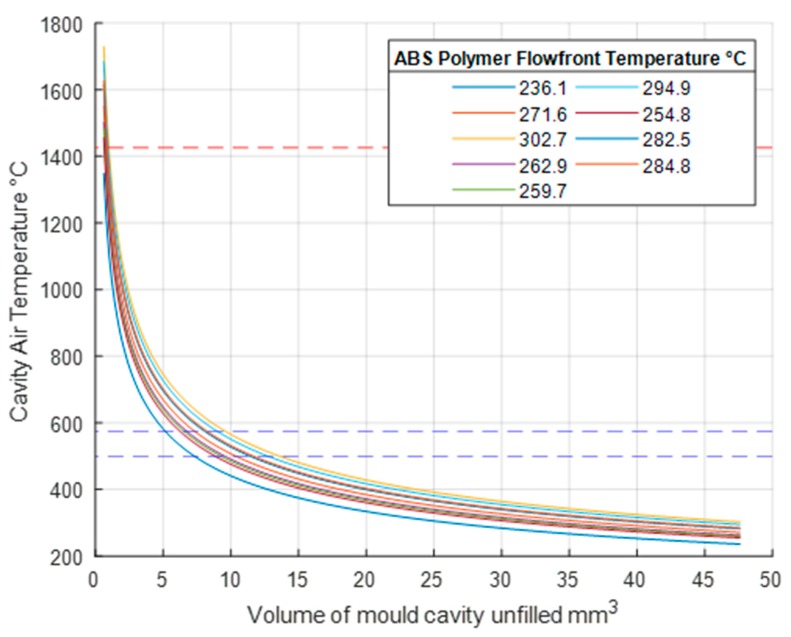
Adiabatic temperature increase when air is compressed in mould, ABS.

**Figure 9 micromachines-11-00358-f009:**
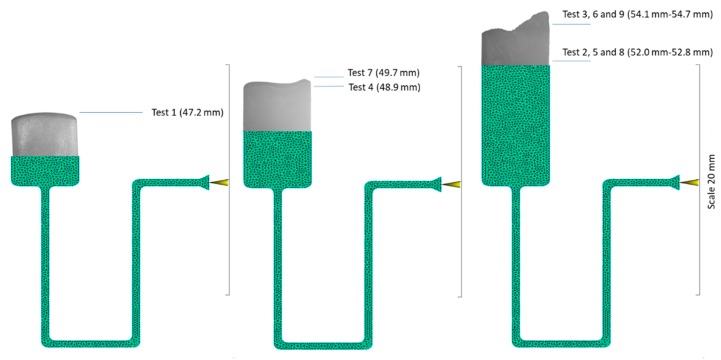
PP flow lengths for physical Tests 1-9.

**Figure 10 micromachines-11-00358-f010:**
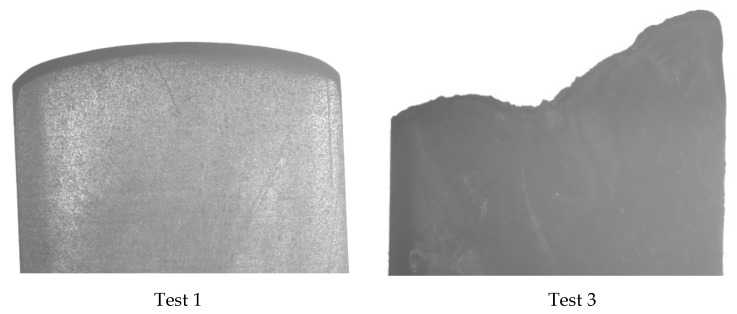
Flow fronts for test parts 1 and 3.

**Figure 11 micromachines-11-00358-f011:**
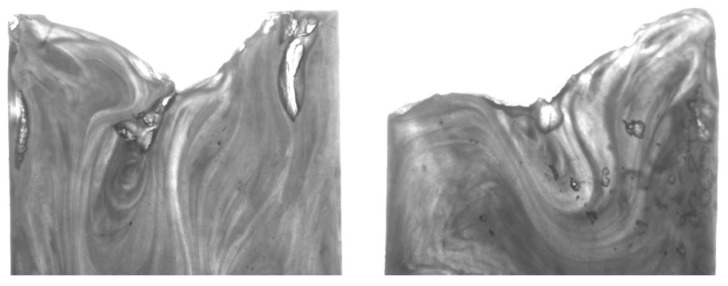
Test part 3 flow fronts with gas trapped within the parts.

**Table 1 micromachines-11-00358-t001:** Section dimensions for the test part.

Length	L1	L2	L3	L4	L5
**Distance (mm)**	5	14.5	7.3	14	15
**Length sum (mm)**	5	19.5	26.8	40.8	55.8

**Table 2 micromachines-11-00358-t002:** Material properties of Polypropylene (PP) and Acrylonitrile butadiene styrene (ABS).

Description	(PP)	(ABS)
Family Name	Polypropylenes (PP)	Acrylonitrile Copolymers
Trade Name	SABIC PP 56M10	MAGNUM 8434
Manufacturer	SABIC Europe B.V.	Trinseo EUR
Moldflow Viscosity Index	VI(240)0087	VI(240)0212
Transition Temperature °C	150	50
**Specific Heat Data**		
Temperature °C	240	240
Specific Heat (Cp) J/kg·°C	2750	2032
**Thermal Conductivity Data**		
Temperature °C	240	240
Thermal Conductivity W/m·°C	0.18	0.152
**Mechanical Properties**		
Elastic Modulus	1340 MPa	2240 MPa
Poisson Ratio	0.392	0.392
Shear Modulus	481.3 MPa	804.6 MPa
**Environmental Impact**		
Resin ID code	5	7
Energy Usage Indicator	3	5

**Table 3 micromachines-11-00358-t003:** Mechanical and thermal properties of P-20 tool steel.

**Mould Specific Heat**	460 J/Kg·°C
**Mould Thermal conductivity**	29 W/m·°C
**Elastic Modulus**	205,000 MPa
**Poisson ratio**	0.29

**Table 4 micromachines-11-00358-t004:** Test parameters.

Test No.	Melt Temp (°C)		Mould Temp (°C)		Injection Speed (mm/s)	
	PP	ABS	PP	ABS	PP	ABS
Test 1	220	220	20	40	200	200
Test 2	250	250	40	60	500	500
Test 3	270	280	60	80	800	800
Test 4	220	220	40	60	800	800
Test 5	250	250	60	80	200	200
Test 6	270	280	20	40	500	500
Test 7	220	220	60	80	500	500
Test 8	250	250	20	40	800	800
Test 9	270	280	40	60	200	200

**Table 5 micromachines-11-00358-t005:** Flow lengths for tests of PP and ABS [30].

Test Number	Flow Length PP (mm)	Flow Length ABS (mm)
Test 1	47.2	32.6
Test 2	52	42.4
Test 3	54.7	45
Test 4	48.9	39.8
Test 5	52.8	32.5
Test 6	54.6	29.3
Test 7	49.7	32.8
Test 8	52.7	36.4
Test 9	54.1	33.1

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
