# Peer review of "High Temperature Adiabatic Heating in µ-IM Mould Cavities—A Case for Venting Design Solutions"

_micromachines, 2020, doi:10.3390/mi11040358_

Round 1

Reviewer 1 Report

line 35: 'transition temperature'

line 37: 'melt flow'

line 44: 'data are'

line 123: correct reference should be inserted

line 199: refer to the used equations

line 232: your citation mentions a chemical degradation. Since you are dealing with thermal degradation there is the question on how is this linked?

line 240: 'dashed lines indicate'

line 250: provide a definition of flow length and the procedure how it is measured (since you are dealing with curved and irregular flow fronts.

line 252: you refer to Table 4: where is it?

line 255: 'therefore'

ad chapter 3.3: you compare simulation and experiment. To judge the quality of the simulations you should add a comparision of the different flow fronts arising from simulation and related experiments based on the used DOE.

line 292: 'gases'

Author Response

Reviewer 1

·         line 35: 'transition temperature'

This is resolved

·         line 37: 'melt flow'

This is resolved

·         line 44: 'data are'

I believe ‘data is’ is the preferred text

·         line 123: correct reference should be inserted

This is resolved

·         line 199: refer to the used equations

This is resolved

·         line 232: your citation mentions a chemical degradation. Since you are dealing with thermal degradation there is the question on how is this linked?

The paper discusses chemical reactions within the polymer brought on by elevated temperatures. We have now added this to the text.

·         line 240: 'dashed lines indicate'

This is resolved

·         line 250: provide a definition of flow length and the procedure how it is measured (since you are dealing with curved and irregular flow fronts.

We have added a description to clarify this (lines 266-269)

·         line 252: you refer to Table 4: where is it?

This was an error and is now resolved

·         line 255: 'therefore'

This is resolved

·         ad chapter 3.3: you compare simulation and experiment. To judge the quality of the simulations you should add a comparision of the different flow fronts arising from simulation and related experiments based on the used DOE.

We have modified this section with an added figure included.

·         line 292: 'gases'

This is resolved

Reviewer 2 Report

The authors have resolved most of the comments made on the previous version.

Above all, the experimental part has been significantly extended.

However, some clarification is needed on this new experimental part in order to fully understand the proposed figures.

Why did the trial only involve PP and not ABS? They are two very different materials and they work at different temperatures with different HTC.

As far as experimentation with PP is concerned, it would be useful to know what flow rate is needed to completely fill the component and it would be advisable to add a photo of the fully filled component.

In addition, it would be advisable to have next to the images of figure 9 proposed, also the images of the relative complete component, as well as the magnification on the proposed area of interest so as to be able to understand how much has been filled the component that instead is not decipherable.

Also for the images of Figure 10, it might be useful to know what magnification we are talking about and, compared to the picture of figure 9, which part of the component has been the object of this analysis and therefore which part has been enlarged.

Figures 4 and 5 are now more complete with the addition of a clearer and more detailed legend.

I still have my doubts about the length of the power supply which, in my opinion, only increases the airflow inside the main cavity, but at this point I think it is wanted.

Even the adiabatic model, in my opinion, is not reproducible for the reasons already explained in the previous revision, that is for the presence of game to the extractors and the interface between the two mould plates.

Author Response

Reviewer 2

The authors have resolved most of the comments made on the previous version. Above all, the experimental part has been significantly extended. However, some clarification is needed on this new experimental part in order to fully understand the proposed figures.

  • Why did the trial only involve PP and not ABS? They are two very different materials and they work at different temperatures with different HTC.

As mentioned in the text (line 272) the ABS physical part failed to fill and because of this we could not witness a gas trap so we removed these results. If we modified the mould dimensions to facilitate easier filling a gas trap would be expected.

  • As far as experimentation with PP is concerned, it would be useful to know what flow rate is needed to completely fill the component and it would be advisable to add a photo of the fully filled component.

The filling is a dynamic relationship, however for this particular design the melt temperature was dominant. But the highest flow was achieved with a high temp and the highest flow rate of 800 mm/s. We have now added additional text in section 3.3 to clarify this (lines 275-277)

  • In addition, it would be advisable to have next to the images of figure 9 proposed, also the images of the relative complete component, as well as the magnification on the proposed area of interest so as to be able to understand how much has been filled the component that instead is not decipherable. Also for the images of Figure 10, it might be useful to know what magnification we are talking about and, compared to the picture of figure 9, which part of the component has been the object of this analysis and therefore which part has been enlarged.

An additional Figure (figure 9) has been added to provide further clarity on the part filling. Also an additional Table (table 1) has been added to provide father clarity on the part dimensions and flow length. We have also modified Figure 1 to support this table.

  • I still have my doubts about the length of the power supply which, in my opinion, only increases the airflow inside the main cavity, but at this point I think it is wanted.

I think in this case you mean the runner. We are confident that there cannot be an extra introduction of air as the resident air has a fixed value. The design of the runner was specifically chosen because it will introduce shear (as shown) and that this shear should increase the temperature of the polymer leading to a potential adiabatic heating. This is precisely the effect that we wanted within the cavity.

  • Even the adiabatic model, in my opinion, is not reproducible for the reasons already explained in the previous revision, that is for the presence of game to the extractors and the interface between the two mould plates.

The PP results show that there is clear evidence of gassing within the mould and damage to the parts. These results coupled with our moulding experience and knowledge of venting challenges form the basis of our hypothesis. Micro moulds can be closed systems which is why venting needs to be a mould design priority if tool life and part quality are to be maintained.

Regarding the adiabatic model, this is a good point. Our hypothesis is that venting rules for macro moulding cannot be easily adapted to micro moulding. Some vents for some polymers are larger than micro components and the rules for split lines and ejector clearances cannot be applied. We failed to sufficiently make this argument in the original text so we have now modified section 1 (lines 78-96) to address this

Regarding the ideal adiabatic model, yes this is the case that it is ideal in the sense that it can be used for zero venting. The worst-case scenario of reaching 1600 DegC is unlikely as this would be with zero venting. This is why we applied a volume of the mould cavity range on the axis in figure 7 and 8. This range helps us understand the amount of unfilled volume of air in the cavity and can thus relate to the amount of air that is escaping. In micro moulding the injection speed is far in excess of macro moulding, and due to the accuracy of the moulds the vents options are much less. What we can show from out model is that extremely high temperatures can be realized and these are far in excess of the flashpoint for the two polymers. So even if there is venting, the model shows that high temperatures are present and they can damage the parts and moulds. Experience has shown us that macro moulds with blocked vents and micro moulds with a lack of venting do get damaged frequently and I think this paper supports the need for a design approach to minimize the phenomena of gassing.

Round 2

Reviewer 2 Report

The proposed revision and the reply of the authors clarified further doubts and improved the overall quality of the article.
However, in my opinion, further clarification is needed on Figure 9. Frankly, the image is not clear to me and also the written text at lines 270-278 is not enough really useful to clarify Figure 9. What is represented with green mesh? The simulated result? and the larger grey mesh is the experimental part? If this is the case, the simulator underestimates the filling in a clear way from the qualitative point of view and, in my opinion, also from the quantitative point of view. And also the sentence written in the conclusions "Autodesk Moldflow simulations of an established part design can predict accurate shear rate and temperature distributions within the μ-IM process" at this point is not well supported.

Author Response

  • The proposed revision and the reply of the authors clarified further doubts and improved the overall quality of the article. However, in my opinion, further clarification is needed on Figure 9. Frankly, the image is not clear to me and also the written text at lines 270-278 is not enough really useful to clarify Figure 9. What is represented with green mesh? The simulated result? and the larger grey mesh is the experimental part?

Figure 9 shows the CAD of the part and it is meshed, it was not possible to take a photo of the whole part so we used the CAD and the 3 end of fill images imposed on top of the CAD to help identify the different flow lengths. I changed the figure title to add clarity to this ‘PP flow lengths for the physical experiments 1-9’

Regarding the supporting text we have added further clarification in relation to the figure and this is highlighted in the text.

  • If this is the case, the simulator underestimates the filling in a clear way from the qualitative point of view and, in my opinion, also from the quantitative point of view.

The aim of the paper is not to compare simulation and physical experiments in terms of the flow length, we have written several journal papers and looked at different simulation models in this area but this submission has a different focus. The end of fill differences between simulation and physical experiments for this particular test part will be different because the simulation code does not allow for adiabatic heating (this would be interesting research) and it also does not account for gassing as the model assumes full venting (we have now added this point to the text after Line 278). The purpose for running the simulation is that it gives us a strong indication of the temperature of the polymer in transit. Then we can use this temperature data and use it as a boundary condition for the adiabatic model. Our other option here would be to use thermocouples within the mould cavity to validate the simulation temperature. And then add this data to the adiabatic model. But we believe that sufficient research has been conducted by ourselves and others in Moldflow simulation at the microscale to support the temperature findings.

  • And also the sentence written in the conclusions "Autodesk Moldflow simulations of an established part design can predict accurate shear rate and temperature distributions within the μ-IM process" at this point is not well supported.

Regarding the shear rate yes this is a good point and we will remove this from the text, however the use of a DOE has given us a very good indicator of the melt temperature during processing and the temperature findings are in alignment with existing research so we will stand by this statement.

This manuscript is a resubmission of an earlier submission. The following is a list of the peer review reports and author responses from that submission.

Round 1

Reviewer 1 Report

This study is to investigate the adiabatic compression and heating of the trapped air during a melt filling stage, which are likely to occur under high setting conditions of process parameters for flowability in microinjection molding. The heating of the air can result in the diesel effect causing deterioration of the mold wall as well as the polymer material. For the mold cavity and the 40 mm-long runner of 0.5 mm thickness, MoldFlow filling simulations were performed using PP and ABS materials. Flow front temperatures and shear rates according to three-level conditions of melt temperature, mold temperature and injection speed were shown and temperature increases of the compressed air were also calculated.

Excessive temperature rise of the adiabatically compressed air is considered to be an important issue. In most case, the air vent needs to be installed in a mold. Even if the air vent is absent, the diesel effect is not easy to predict and it is not so convincing whether the temperature will reach so high temperature over 1000 Celcius degree because different assumptions may be made. The molten polymer could be compressible under high pressure conditions. Also, the mold cavity on the parting line is not thought to be a complete closed system when considering the flashes along the parting planes observed in other studies.

Most of all, this study lacks experimental verification and entirely depends on simulation results under a simplified assumption. It can be a possible explanation on the effect of fast filling conditions but doesn't provide systematic approach to the venting design solution.

Reviewer 2 Report

The proposed article deals with a rather controversial topic in the field of injection molding that in the micro molding field can potentially be a problem in terms of accuracy for the production of a final component.

However, there are some aspects that I think should be reviewed to improve their quality.

First of all, the title speaks about “Venting design solution”, but in my opinion the proposed component is not optimal for this type of study. The length of the runner, in my opinion, involves an extra introduction of air into the cavity, in addition to forcing the polymer to run for a long time before reaching the real cavity, probably introducing many other aspects that can increase the temperature (surface roughness for example).

Moreover all the work is focused on the simulative approach, without a concrete comparison with the experimental aspects.

Furthermore, it is not clear what is the point considered for the temperature measurements introduced in Figure 4 (where is?) .

Further doubts arise from the use of the adiabatic model, which in its ideal version could also lead to an excessive rise in trapped air and forced to compress itself to create space for the polymer melt. In the real case, however, this could never happen for many reasons: the presence of extractors that inevitably have a tolerance that is not too narrow, which guarantees the evacuation of the air, as well as the plane of separation between the two plates, the heat transmission which in micro injection is very fast and often causes fast cooling, etc ...

I also found it difficult to clearly follow the explanation of the two graphs in Figure 3 (the legend must be completed) and 5.

Also all the analysis, in my opinion too long, on the possible overheating up to almost melting temperature of the steel that ideally could reach the air when in contact with a high temperature fluid in a rather narrow space, is an unrealistic forcing in reality.

Beside this, a careful reading of the text should be done since some sentences need to be reviewed in the english formto imporve clarity, there are several references to skipped figures, Figure 4 is missing but is cited and you can repeatedly find "diesel affect" instead of "diesel effect".

The paper should be reconsidered after major revisions

English language need to be imporved due to minor errors.

Reviewer 3 Report

The paper discuss the need for vacuum venting in micro injection molding. Simulations are carried out to evaluate the effect of process parameters on polymer temperature, and shear rate. Please see my comments below.

English style and grammar should be checked throughout the text. Introduction, Lines 28-44. When describing the effect of different process parameters, the response variable selected for the analysis should be mentioned. Section 2.0 - considering the Design proposed for the part, the literature review reported in the Introduction might not be very significant. Most of the papers that the Authors reported refer to micro molding of micro/nano-structured surfaces. Venting of the geometry indicated here is easy solved with basic injection mold design solutions. Line 132, Line 143-144. It looks like there was a reference error here. Figure 3 (Line 166), should be Figure 4. Figure 3 (Line 166), Figure 5. I recommend using a different type of plot, or using a physical quantity in the x-axis. The continuous variations between the different tests are confusing.  Lines 171-173. `The 172 findings demonstrate that the process parameters have a major influence on the resulting shear but 173 importantly no single test goes above the critical shear rate for PP (1x105 1/S) and ABS (5x104 1/S).` How have the Authors characterized these values?